# TABULAR DATA IMPUTATION
# CHOOSE KNN OVER DEEP LEARNING

## ABSTRACT

Recent infatuation for Artificial Neural Networks have led to the development of complex and powerful algorithms for data imputation. This study is compares the best state-of-the-art deep-learning models with the well-established KNN algorithm (1951). By using real-world and generated datasets in various missing data scenarios, we show that the good old KNN algorithm is still competitive (nay better) than powerful deep-learning algorithms for tabular data imputation. This work advocates for an appropriate and reasonable use of machine learning.

## 1 INTRODUCTION

Missing values constitute a serious issue in data science. Incomplete datasets result from uncollected, lost or voluntarily deleted observations. Most machine learning algorithms cannot handle datasets with missing values, and meticulous data preprocessing is therefore needed. The most common preparatory processing consists of Listwise deletion, where only complete observations are kept for further data analysis. This solution, although straightforward to implement, has two main disadvantages: it significantly reduces the size of the dataset and induces a bias, which may lead to false positive research claims (Lall (2016)).

Data imputation is an alternative preprocessing method which involves estimation and replacement of the missing values. It allows to preserve the whole dataset for subsequent analysis but requires careful handling as it can also introduce a bias in the imputed dataset (Gelman & Hill (2006)). In particular, only "well-behaved" values can be imputed and missing values of outlier observations will get overlooked (Binder (1996)).

Several data imputation algorithms have been proposed over the last 70 years: Hot-deck, Cold-deck, Mean-value substitution (Kalton & Kasprzyk (1986)) , C4.5 algorithm (Salzberg (1994)), CN2 induction algorithm (Clark & Niblett (1989)), KNN (originally developed in 1951 - Fix & Hodges (1989) - and later expanded by Altman (1992)). These traditional machine learning algorithms have been shown useful in the past. Notably, KNN has shown best imputation quality amongst these various data imputation algorithms (Batista & Monard (2002), Bertsimas et al. (2018), Poulos & Valle (2018), Jadhav et al. (2019)).

Over the last decade, Artificial Neural Networks (ANN) have transformed the fields of statistics, computer science and data analysis. More precisely, Generative Adversarial Networks (GANs) have shown spectacular results at generating new observations from a given data distribution. As such, GAN models have been developed to tackle the problem of missing values in data science. GAIN (Yoon et al. (2018)) and MisGAN (Li et al. (2019)) claim state-of-the-art data imputation results. While GAIN has been specifically developed for tabular data imputation, MisGAN is primarily designed to impute degraded images and can be adapted to work with tabular data.

This work intends to compare data imputation performances of state-of-the-art GAN models (GAIN and MisGAN) with the standard already established K-nearest neighbours algorithm (KNN). We evaluate the imputation quality in MCAR, MAR and MNAR scenarios (see Section 2). The present study is restricted to tabular data, that is numerical data we can arrange into rows and columns in the form of a table with cells.

Section 2 explains the various missing data scenarios and their properties. Section 3 presents the algorithms used in this work. Section 4 introduces the datasets. Section 5 details the method of this

study. Section 6 shows the data imputation results. Section 7 includes a summary, a discussion and final recommendations.

## 2 MISSING DATA SETTINGS

Let $x$ denote the complete vector of a given observation and $m$ its missing value binary mask, so that the actual available data is the element-wise product $\widetilde{x} = x \odot m$. Data imputation methods seek to estimate the missing values of $\widetilde{x}$ by taking advantage of patterns in the dataset and using observed properties of $\widetilde{x}$. The probability distribution of $m$ is referred to as the *missing data mechanism*. Following the classification established by Little & Rubin (2014), missing data scenarios belong to one of the following three settings.

### 2.1 MISSING COMPLETELY AT RANDOM (MCAR)

In MCAR scenarios, the missing data mechanism is assumed to be independent of the intrinsic probability distribution of $x$, and occurs completely at random, such that $p(m|x) = p(m)$. We can generate MCAR missing values in a complete dataset by hiding each cell with a given probability, for example 10%. There exist statistical tests to assess whether the MCAR assumption holds or not (Little (1988)).

Apart from the obvious information loss, MCAR scenarios are convenient assumptions. Indeed, the subset of complete observations is free of bias, and we may therefore safely proceed with further data analysis techniques. However, MCAR is often an unrealistically strong assumption in practice.

### 2.2 MISSING AT RANDOM (MAR)

MAR scenarios occur when the missing data mechanism can be fully explained by the observed data. We may write $p(m|x) = p(m|\widetilde{x})$. For example, males are less likely to complete a depression survey, regardless of their depression level. As long as the gender of every participant is known, the dataset can be treated as MAR by assuming that missingness only depends on maleness.

MAR assumption is more general than MCAR, but cannot be statistically tested and one must rely on its fairness. Subsequent statistical analyses may include biases which can theoretically be corrected under the MAR assumption.

### 2.3 MISSING NOT AT RANDOM (MNAR)

Finally, MNAR scenarios encompass every other scenario: the reason why data is missing depends on unobserved variables. In the previous example, this is the case if respondents do not fill the depression survey because of their actual depression level, or because of another unrecorded variable like their income for instance. As MCAR and MAR are convenient assumptions to facilitate the treatment of incomplete datasets, most missing data mechanisms are actually distributed in a MNAR setting.

While being truthful, MNAR scenarios have no systematic ways to handle missing values. One strategy is to accumulate more data in the hope of understanding the missing data mechanism.

## 3 DATA IMPUTATION ALGORITHMS

We select, analyse and compare the performances of the best algorithms to date. While some tabular data imputation methods learn a probability distribution on a complete dataset before performing data imputation, we focus on algorithms which use the entire dataset (including missing values) to perform in-place imputations. On the one hand, GAIN (Yoon et al. (2018)) and MisGAN (Li et al. (2019)) are new state-of-the-art ANN generative models tailored for data imputation. On the other hand, the KNN algorithm is a non-parametric method which can be used for data imputation.

## 3.1 PRESENTATION OF GAIN

Generative Adversarial Imputation Nets (GAIN) have been proposed in 2018 as a generative model specifically designed for tabular data imputation problems. GAIN generalises the well-established architecture of GAN models by looking at individual cells rather than complete rows. The authors claim state-of-the art imputation quality performances (Yoon et al. (2018)).

Given an incomplete observation, GAIN works as follows:

1. Cells with missing values are filled by random noise while observed values are kept unchanged. A binary mask telling whether each cell is genuine or comes from the random noise is concatenated to the observation, therefore doubling its length.

2. This vector is provided to the GAIN generator which outputs a generated vector. The original data vector is imputed by conserving genuine cells and replacing missing cells by their corresponding values in the generated vector.

3. The imputed vector is presented to the GAIN discriminator which tells apart genuine values from imputed values. The GAIN discriminator works on a cell-by-cell basis instead of considering the row as a whole. Both GAIN generator's and discriminator's parameters are trained following the standard adversarial process used by GAN models.

GAIN's parameters are updated to minimise the binary cross-entropy loss function for the discriminator given by

$$\mathcal{L}_D(i) = -\sum_{j, b_{ij}=0} m_{ij} \log(\widehat{y_{ij}}) + (1 - m_{ij}) \log(1 - \widehat{y_{ij}})$$

where $m_{ij}$ is 0 if cell $(i, j)$ is missing or 1 otherwise, $b_{ij} = 0$ indicates cells that have not been hinted to the discriminator, and $\widehat{y_{ij}}$ is the discriminator prediction for cell $(i, j)$; and a combination of binary cross-entropy (for the generated cells) and RMSE for the discriminator given by

$$\mathcal{L}_G(i) = \mathcal{L}_{G_1}(i) + \alpha \mathcal{L}_{G_2}(i)$$

with     $\mathcal{L}_{G_1}(i) = -\sum_{j, b_{ij}=0} (1 - m_{ij}) \log(\widehat{y_{ij}})$     and     $\mathcal{L}_{G_2}(i) = -\sum_j m_{ij}(\tilde{x_{ij}} - x_{ij})^2$

where $\tilde{x_{ij}}$ is the generator output and $x_{ij}$ the actual observation for cell $(i, j)$.

GAIN has two hyperparameters to manually tune: the hint fraction $h$ of true labels to provide the discriminator to stabilise learning, and $\alpha$ the relative weight between cross-entropy and RMSE for the generator loss. After several trails, we decide to use the recommended default values of Yoon et al. (2018). Besides, the most relevant hyperparameter for GAIN is the number of training epochs, which we discuss in Section 5.

## 3.2 PRESENTATION OF MISGAN

MisGAN has been introduced in 2019 as another GAN model framework capable of handling complex datasets with missing values. Primarily developed for image completion, it can be adapted to handle tabular data. MisGAN claims state-of-the-art imputation quality for images (Li et al. (2019)).

In its original design, MisGAN is composed of three pairs of generator and discriminator. A first pair $(G_x, D_x)$ attempts to model the probability distribution of the data, while another pair $(G_m, D_m)$ tries to model the missing data mechanism. Both generators $G_x$ and $G_m$ are used in parallel to produce fake deteriorated observations. Finally, a third pair $(G_i, D_i)$ is used to perform the imputation of missing values.

When adapted to tabular data, the missing data mechanism has a simple distribution and does not require complex neural networks to be modelled. Instead, we choose to draw missing masks directly from the observations in the dataset.

Like for GAIN, the performances of MisGAN mostly depends on the number of training epochs. Section 5 explains how we fix the number of training epochs for MisGAN. The artificial neural network architecture of GAIN and MisGAN are both presented in Appendix A.

### 3.3 PRESENTATION OF KNN

The K-Nearest Neighbours algorithm is a non-parametric method, originally developed for classification tasks in 1951 (Altman (1992)). For data imputation tasks, the KNN algorithm selects the $K$ nearest neighbours of a given incomplete observation, and uses available data from the selected neighbours to estimate missing values. Despite its simplicity and its age, the KNN algorithm has been shown to outperform traditional data imputation algorithms (Batista & Monard (2002)).

KNN imputes missing values using a weighted average of the selected neighbours. The most two common weighting systems are `uniform` and `distance`, respectively defined by $\frac{1}{K}$ (where $K$ is the number of neighbours) and $\frac{1}{d_{ij}}$ (where $d_{ij}$ is the distance between observations $i$ and $j$. We respectively refer to these algorithms with the names KNN-uniform and KNN-distance.

## 4 DATASETS

Real-world and well-behaved simulated datasets are both used in this comparative study. The real-world datasets are taken from the open access UCI Machine Learning Repository (Dua & Graff (2019)). For the sake of reproducibility, we make use of the same datasets used by authors of GAIN (Yoon et al. (2018)). And for the sake variety, two additional datasets have been selected: white wine and red wine datasets. Table 1 shows the size of the nine datasets.

We insist that a comprehensive understanding of the data at play is the most important preliminary step before any data analysis. Accordingly, Appendix B provides all essential details about the data used in this work. It is worth noting that all following datasets are originally complete, i.e. they do not have missing values in the first place.

## 5 METHODS

This section describes the methodology of this work. We begin with finding the appropriate hyperparameters of the algorithms used. Then, we conduct several experiments with varying missing rates, missing data scenarios and datasets. Throughout the following experiments, we first inject missing values and immediately scale the datasets in the interval [0, 1]. After imputation, the performances are computed using the RMSE between ground truth and imputed missing values directly on the scaled variables. Each experiment is repeated 20 times to allow for the computation of the mean and the standard deviation of the RMSE reported in each subsequent plot.

### 5.1 TUNING THE HYPERPARAMETERS

We use the generated mixture of three Gaussians dataset (c.f. `mydata2` dataset, Appendix B.7) in MCAR scenario with 20% missing rate to select appropriate hyperparameters for the various algorithms of this comparative study. For the following hyperparameter search, we first introduce missing values and then use the algorithms with their pre-selected hyperparameters values and repeat this experiemtn 20 times. We report the mean and standard deviation of the imputation RMSE in Appendix C to select the best hyperparameter values.

GAIN and MisGAN need the number of training epochs to be specified. We train from 1,000 to 20,000 epochs by steps of 1,000 epochs and choose the number of training epochs which minimises the imputation RMSE (see Appendix C.1): 20,000 epochs for GAIN and 5,000 epochs for MisGAN. MisGAN becomes unstable after 7,000 epochs. As GAIN performances keep decreasing, we train with more epochs (see Appendix C.2) and decide to stick with 20,000 epochs.

For KNN, the hyperparameter is the number of neighbours $K$. We use from 2 to 300 neighbours by steps of 2, and select $K = 50$ neighbours to minimise the imputation RMSE for both weight systems (see Appendix C.3).

Note that these hyperparameters may strongly depend on the dataset size. The dataset used for hyperparameters tuning has 1,000 rows. For following experiments, we compute a multiplicative factor to preserve similar proportions $f = \frac{n}{1000}$, where $n$ is the number of rows in the current dataset. The number of training epochs accordingly becomes $\frac{5000}{f}$ for MisGAN and $\frac{20000}{f}$ for GAIN, and

the number of neighbours is $50f$ for both KNN algorithms. As an example, a dataset two times larger will use twice as many neighbours and half the number of training epochs. We experimented this approach on two datasets: `wine_white` and `breast`. The former has about five times more entries than `mydata2` dataset, while the later has about half. Appendix C.4 discuss the reliability of the hyperparameter selection tuning process.

## 5.2 EXPERIMENTS

The following experiments are conducted 20 times to compute the mean and standard deviation of the imputation RMSE. Results and interpretation are provided in the next section.

1. Varying the missing rate in MCAR setting from 10% to 80% by steps of 10%, with the generated mixture of three Gaussians dataset.

2. Performances on real-world datasets in MCAR setting with 20% missing rate.

3. Performances over all datasets in MAR setting, with overall 20% and 45% missing rates.

4. Performances over all datasets in MNAR setting, with overall 20% and 45% missing rates.

To generate MAR missing values, we select a column that we keep untouched throughout the full MAR experiment. We compute the quantiles of the selected column, scale the quantiles between $0$ and $2\mu$ (where $\mu$ is the overall missing rate) and interpret these values as common missing rates for all other columns. For example, in MAR setting with an overall 20% missing rate, the quantiles of the selected column are scaled in the range $[0\%, 40\%]$. The selected variable for each dataset along with its meaning are reported in Table 1.

**Table 1:** List of the selected variables for MAR settings. The datasets' sizes are also provided as well as the column number of the variable used to generate MAR missing data.

| Dataset name | Size | MAR column | Meaning |
|---|---|---|---|
| `breast` | (569, 30) | 0 | mean cell radius |
| `credit` | (30000, 14) | 1 | customer's age |
| `letter` | (20000, 16) | 4 | total number of black pixels |
| `news` | (39644, 44) | 0 | number of words in the title |
| `spam` | (4601, 57) | 54 | average length of uppercase letters |
| `wine_red` | (1599, 12) | 0 | fixed acidity |
| `wine_white` | (4898, 12) | 0 | fixed acidity |
| `mydata1` | (1000, 5) | 0 | No meaning |
| `mydata2` | (1000, 5) | 0 | No meaning |

To generate MNAR missing values, we compute the quantiles for every individual column (in the same way as MAR setting) which we scale in the interval $[0, 2\mu]$. We interpret these values as the missing probability for every corresponding cell. In MNAR settings, the higher the value of a cell relatively to its column, the more likely it will be missing. Looking at the first column of the mixture of three Gaussian dataset for example, data from the orange cluster is more likely to be missing than data from the blue cluster (see Fig. 5 in Appendix B).

## 6 RESULTS AND INTERPRETATION

### 6.1 VARYING THE MISSING RATE IN MCAR SETTING

The first experiment consists in changing the missing rate from 10% to 80% (by steps of 10%) on the generated mixture of three Gaussians dataset in MCAR setting (`mydata2`). The imputation results are shown in Fig. 1 (a). As the RMSE for MisGAN is comparatively large, we only display the results of GAIN and KNN for clarity (for complete results and discussion on MisGAN, see Fig. 11 in Appendix D.1). We decide not to show the results of MisGAN from now on (they are provided in the supplementary materials, in Appendix D.1). For numerical values, see Tables in Appendix G.

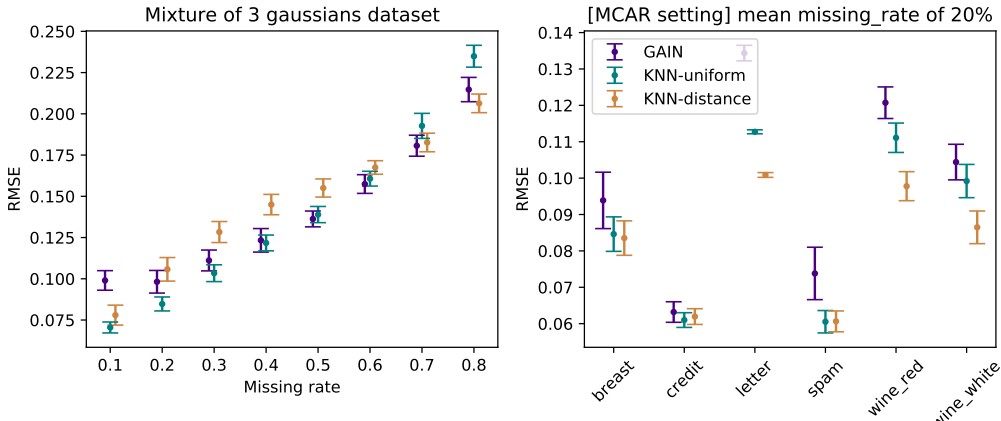

**Figure 1:** MCAR scenario experiments. (a) Left panel: imputation RMSE for the mixture of 3 gaussians dataset (`mydata2`) with varying missing rates. (b) Right panel: imputation RMSE for all real-world datasets (except the `news` dataset).

Not surprisingly, the performances of GAIN and KNN deteriorate with increasing missing rates. GAIN remains competitive with both KNN algorithms. It is worth noting that KNN-uniform performs better at low missing rates, while KNN-distance performs better at high missing rates. Intuitively, the weight system provides KNN-distance with better resilience to bad neighbours (which are more common with high missing rates).

## 6.2 REAL-WORLD DATASETS IN MCAR SETTING

The next experiment uses real-world datasets in MCAR setting with 20% missing rate. Fig. 1 (b) shows the imputation quality results, where we can see that both KNN algorithms perform better than GAIN over all available datasets. From now, we also decide not to report the RMSE for the `news` dataset since the performances are chaotic regardless of the imputation method (for explanation, discussion and the complete results, see Fig. 12 in Appendix D.2).

We could not reproduce GAIN results. Intriguingly, the imputation of GAIN in this work outperforms the imputation from the original authors on the `credit` dataset: RMSE of 0.063 here, and 0.186 in Yoon et al. (2018). For the other datasets, the performances of GAIN in this work are slightly lower than the original authors' reported RMSE. Exhaustive comparison is available in Appendix E.

## 6.3 MAR EXPERIMENTS

This section refers to Fig. 2. With a missing rate of 20% in MAR setting (left hand panel), both KNN algorithms perform slightly better than GAIN over all datasets. When we increase the missing rate to 45% (right hand panel), we see that the imputation quality overall decays. The performances of GAIN, KNN-uniform and KNN-distance are now comparable, with sometimes one method significantly performing better. We notice that the trends and orders of magnitude are being preserved across datasets when the missing rate increases. Look at the `breast` dataset for instance: it has a high RMSE with GAIN both with 20% or 45% missing rate data.

## 6.4 MNAR EXPERIMENTS

This paragraph refers to Fig. 3. In MNAR setting with a missing rate of 20% (left hand panel), there is no clear best imputation method. All imputation RMSEs remain below 0.15 on average, and are very close to the imputation RMSEs obtained in the MAR setting (cf. Fig. 2 (a)). With a missing rate of 45% (right hand panel), the RMSE becomes even larger than with the MAR experiments. We can argue that MNAR missing values are harder to recover than MAR missing values, since the missing data mechanism hides itself in the process of generating missing values. GAIN performs

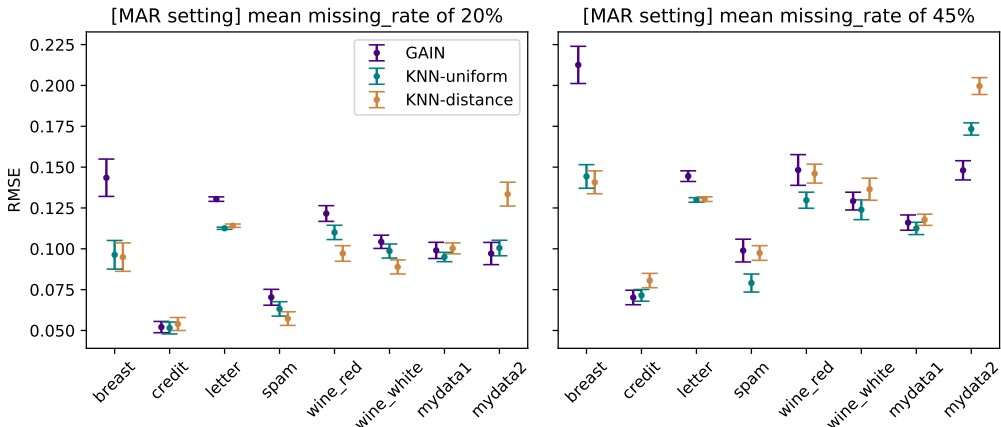

**Figure 2:** MAR scenario experiments for all datasets (except `news` dataset). (a) Left panel: MAR setting with average missing rate of 20%. (b) Right panel: MAR setting with average missing rate of 45%.

better imputations than KNN in MNAR setting with 45% missing rate. For the large variance in the `spam` dataset RMSE, see Appendix D.2.

It is worth noting that KNN algorithms perform poorly against GAIN with the generated Gaussians mixture dataset, on which we benchmarked the hyperparameters. The mixture of three Gaussians dataset indeed suffers a lot from the strong case of MNAR mechanism using quantiles, as clusters with large value variables will undergo higher missing rates than clusters high low value variables. Consequently, some clusters concentrate most of the missing data and become challenging to impute with KNN as they have few neighbours with complete observations.

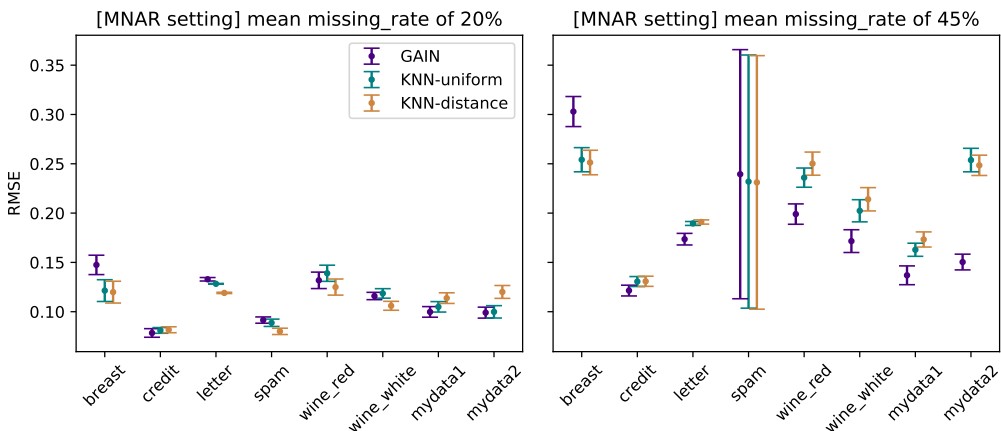

**Figure 3:** MNAR scenario experiments for all datasets (except `news` dataset). (a) Left panel: MNAR setting with average missing rate of 20%. (b) Right panel: MNAR setting with average missing rate of 45%. For the high variability in the imputation RMSE of the `spam` dataset, see Appendix D.2.

# 7 SUMMARY, DISCUSSION AND MESSAGE

## 7.1 SUMMING UP

MisGAN was initially designed to impute missing rectangular pixel blocs in images. While Mis-GAN claims state-of-the-art results for image imputation, this framework does not perform well for tabular data imputation, with large confidence intervals in its predictions. That being said, MisGAN can remain competitive on well behaved datasets with low missing rates. GAIN has been developed

for tabular data imputation and shows good imputation quality, especially in MNAR settings with high missing rates. Like most ANN models, tuning the hyperparameters of MisGAN and GAIN is key to avoid over-training or under-training. Unlike lazy learning algorithms like KNN, Generative Adversarial Network architectures can learn complex relationships between variables. This feature can explain the performances of GAIN in MAR and MNAR settings, where the network could find meaningful patterns in the missingness.

MCAR is an assumption that can be tested (as opposed to MAR assumption). Little's Test of Missing Completely At Random (Little (1988)) can be used to assess whether the MCAR hypothesis holds. The MAR and MNAR missing data scenarios used in this study are strong cases of dependence between the data distribution and the missing data mechanism. Lighter dependencies exist (see supplementary materials for GAIN, Yoon et al. (2018)). Choosing a missing data mechanism is subjective (except for MCAR) and obviously affects the subsequent data imputation quality.

This study shows that the use of massive deep-learning algorithms are not justified for tabular data imputation, and reinforces previous results: Bertsimas et al. (2018), Jadhav et al. (2019), Poulos & Valle (2018), Jäger et al. (2021). Tabular data imputation problems do not require complex machine learning algorithms as it was already established by Batista & Monard (2002).

Lastly, it appears that the imputation quality of an algorithm rather depends on the dataset itself than on the missing data mechanism per se: see the `breast` dataset where KNN systematically performs better than GAIN, and `mydata2` (mixture of three Gaussian dataset) where GAIN performs better than KNN most of the time. For this reason, we recommend to start any data processing with a thorough investigation of the dataset at play, like we show in Appendix B.

## 7.2 DISCUSSION ON MACHINE LEARNING

The development of ANN along with "Deep Learning" is unquestionably an important milestone in Data Science. For example, Convolutional Neural Networks (CNNs) have shown outstanding results in image processing where they can outperform most other traditional methods when properly used. But as for tabular data imputation, traditional and well-established statistical methods are not yet obsolete. There is also an inherent limitation in data imputation problems: we are left with what the (observed) data is. Consequently, it is particularly challenging to recover missing outliers.

On one hand, KNN algorithms require a distance matrix between each observation in the dataset to be computed. This step can take a couple of minutes if the dataset is large (see Appendix E). Once computed, we can instantaneously try any desired number of neighbours for the imputation, which make KNN easily to fine tune. Also, KNN is a deterministic algorithm that can be straight-forwardly reproduced and its results are easy to interpret. On the other hand, it takes several minutes to train GAIN or MisGAN (for 20,000 and 5,000 epochs respectively, on a dataset with 1,000 observations). Fitting massive ANN models can become time and resource consuming. In addition, the ANN "black-box" architecture is often criticised. Stochastic imputations can be useful to derive probability distributions, but prevent from unambiguous replicability. Table 2 gives a summary of this comparison. For more details on the computational training time, see Appendix E.

**Table 2:** Comparison and recapitulation

|  | GAIN | MisGAN | KNN |
|---|---|---|---|
| **Image imputation** | No | Yes | No |
| **Tabular data imputation** | Yes | Inconsistent | Yes |
| **Training time (for $\sim$ 1000 observations)** | $\sim$ 190 sec | $\sim$ 80 sec | $<$ 1 sec |
| **Optimal hyperparameter search** | Need to train again from scratch with new hyperparameters | | Exhaustive and instantaneous |
| **Results** | "Black-box" model Stochastic imputation | | Easy to interpret Deterministic imputation |
| **Reproducibility** | Algorithmically complex | | Straightforward |

Advances in Machine Learning tend to showcase several "state-of-the-art" algorithms, although only one may exist by definition. Exhaustive comparative studies allow to share common benchmarks and fairly assess the performances of each method.

Finally, Machine Learning comes alongside with the improvement of computing power, smaller processors and a digital era with huge amount of data. But this era is also associated with over-consumption, resources problems and alarming ecological consequences. Digital sobriety advocates for using the Internet and technologies in a mindful and responsible way. In light of it, we urge researchers in Machine Learning to develop and promote appropriate and reasonable tools, adapted to the data and problem that we aim to address.

### 7.3 TABULAR DATA ANALYSIS RECIPE

Acknowledging and handling missing values is of major importance in data analysis. Since most Data Science algorithms cannot handle missing values, a straightforward solution consists in deleting incomplete observations. We strongly discourage from doing so: the reason why data is missing may tell a lot about an observation, and a missing value could even provide more information than an actual value, if utilised properly.

It goes without saying that a great understanding of the dataset is the most important factor for a successful data analysis. We suggest to follow these steps before any tabular data analysis with missing values:

1. Do not discard incomplete observations.
2. If possible, handle individual missing values by hand.
3. Look for pattern in the missingness, and understand the missing data mechanism.
4. Test whether the MCAR assumption holds. If not, assess (empirically) if the MAR assumption seems valid.
5. Perform the imputation with a suitable algorithm.

KNN imputation is almost instantaneous while deep-learning models take time and energy to be trained. For its simplicity, reproducibility and intuitive interpretation, we encourage to use KNN for data imputation when possible.

### ETHICS STATEMENT

We assure that this review is the authors' own original work and reflect the authors' own research and analysis in a truthful and complete manner. The developers of the algorithms used in this study are appropriately credited.

We also declare no conflict of interest or discrimination bias in this study. This work reflects fairness concerns and responsible research practices. We are grateful to the developers of GAIN for their valuable input regarding the hyperparameters' tuning of their algorithm.

### REPRODUCIBILITY STATEMENT

The code and the data used in this work are provided as supplementary materials in a zip folder. We have used a pseudorandom number generator seed, so that interested readers can faithfully reproduce all the plots of this work. We also welcome readers to change the seed to assess the statistical consistency of our results.

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

# – Supplementary materials –

## A    APPENDIX: GAIN AND MIisGAN ARCHITECTURES

Table 3 and Table 4 present the architecture of both the generator and the discriminator of GAIN and MisGAN models respectively. Following the architecture of GAIN (Yoon et al. (2018)), we adapted the input and output layers to the number $c$ of columns in the dataset.

**Table 3:** Neural network architecture for GAIN

| Layer | Input shape | Output shape | Nb. params |
|---|---|---|---|
| Fully connected | 2c | 64 | 128c + 64 |
| Fully Connected | 64 | 128 | 8320 |
| Fully Connected | 128 | 128 | 16512 |
| Fully Connected | 128 | c | 129c |

**Table 4:** Neural network architecture for MisGAN

| Layer | Input shape | Output shape | Nb. params |
|---|---|---|---|
| Fully connected | c | 128 | 128c + 128 |
| Fully Connected | 128 | 128 | 16512 |
| Fully Connected | 128 | 128 | 16512 |
| Fully Connected | 128 | c | 129c |

## B    APPENDIX: DATASETS DESCRIPTION

Real-world and well-behaved simulated datasets are both used in this comparative study. Seven real-world datasets are taken from the open access UCI Machine Learning Repository (Dua & Graff (2019)). Because an exhaustive comprehension of the data at play is the most important preliminary step before any data analysis, this appendix provides essential details about the data used in this work. Note that all following datasets are complete: they do not show missing values in the first place. The first two columns of Table 1 shows the dataset names and sizes.

### B.1    BREAST CANCER WISCONSIN (DIAGNOSTIC) DATA SET - BREAST

Ten features are computed from a digitised image of a fine-needle aspiration of a breast mass. They describe characteristics of the cell nuclei shown in the image (Bennett & Mangasarian (1992)). Fine-needle aspiration is a diagnostic procedure which consists in sampling and examining cells under microscope (biopsy), in the search for lumps or tumours.

The dataset contains 596 observations and 32 columns. The first two columns are the ID number and the diagnosis of the cell (malignant or benign). The last 30 columns are comprised of the mean, the standard error and the mean of the largest three values of the following ten cell features: radius, texture, perimeter, area, smoothness, compactness, concavity, concave points, symmetry and fractal dimension. These last 30 columns are kept for data imputation comparison.

### B.2    DEFAULT OF CREDIT CARD CLIENTS DATA SET - CREDIT

Default payments of 30000 customers in Taiwan have been recorded over six months. The dataset comprises the following explanatory variables: total amount of given credit, gender, education, marital status, age, past payment records, amount of bill statement and amount of previous statements (Yeh & hui Lien (2009)).

Regarding data imputation purposes, the categorical socio-demographic variables (gender, education and marital status) and the past payment records are removed. In the end, the dataset includes 30000 rows and 14 columns.

### B.3   LETTER RECOGNITION DATA SET - `LETTER`

Black and white images of the 26 uppercase letters from the roman alphabet are produced using 20 different font styles and random distortions. Sixteen features have been computed on the original images such that the resulting dataset comprises 20000 rows and 17 columns, where the first column tells the corresponding letter and is therefore discarded for this study. The following 16 columns contain integer values ranging from 0 to 15, that we treat as continuous. These 16 columns include features like the height and length of the image in pixels, the number of black pixels in the image, the horizontal and vertical average of black pixels or higher statistical moments (Frey & Slate (1991)).

### B.4   ONLINE NEWS POPULARITY DATA SET - `NEWS`

A large number of online news articles published by Mashable have been collected over the course of two years (Fernandes et al. (2015)). The actual articles can be seen been accessing their corresponding URL link provided in the dataset. A series of 58 features have been computed on the news articles. They include 44 continuous variables - like number of words in the title - and 14 binary variables - indicating whether the article has been published a Monday, for example.

The dataset has 39644 rows and 61 columns. Three additional columns have been discarded for data imputation purposes. They include the URL link of the article and the time delay between the news article publication and the data acquisition (both of which are non-informative) as well as the number of shares of the news article (which is the original target variable).

### B.5   SPAMBASE DATA SET - `SPAM`

Features are extracted from both spam and non-spam e-mails. The dataset comprises 4601 observations and 58 columns corresponding to the following attributes. The first 48 attributes are the frequency (in percent of words in the e-mail) of preselected words like "will", "you" or "free". The next six columns are the frequency (in percent) of the following preselected characters: ';', '(', '[', '!', '$', '#'. The next three columns show the average length of uninterrupted sequence of uppercase letters, the length of the longest uninterrupted sequence of uppercase letters, and the total number of uppercase letters in the e-mail respectively. Finally, the last column indicates whether the corresponding e-mail was considered spam or not and has been discarded for data imputation purposes.

### B.6   WINE QUALITY DATA SET - `WINE_RED` AND `WINE_WHITE`

Various variables have been collected for red and white variants of the Portuguese Vinho Verde wines (Cortez et al. (2009)). The red wines dataset contains 1599 observations while the white wines one has 4989 observations. Both datasets have 12 columns corresponding to the following attributes: fixed acidity, volatile acidity, citric acid, residual sugar, chlorides, free sulfur dioxide, total sulfur dioxide, density, pH, sulphates, alcohol and a final score between 0 and 10 for the wine quality.

### B.7   GAUSSIAN DATASETS - `MYDATA1` AND `MYDATA2`

Two Gaussian datasets have been generated for comparison. We referred them as "well behaved" datasets, in the sense that they do not show outliers as the other seven real-world datasets. Also, their intrinsic probability distributions are perfectly determined. Both Gaussian datasets have 1000 rows and 5 columns. Correlation matrices with high coefficients have been obtained by using random factors (see Appendix F).

The first Gaussian dataset comprises a single multivariate Gaussian distribution (see pairplot, Fig. 4). The second Gaussian dataset has been generated with a mixture of 3 Gaussians from independent correlation matrices obtained with with random factors (see pairplot, Fig. 5). The first class (in blue) has 150 observations, the second class (in orange) has 300 observations and the third class (in green) has 550 observations.

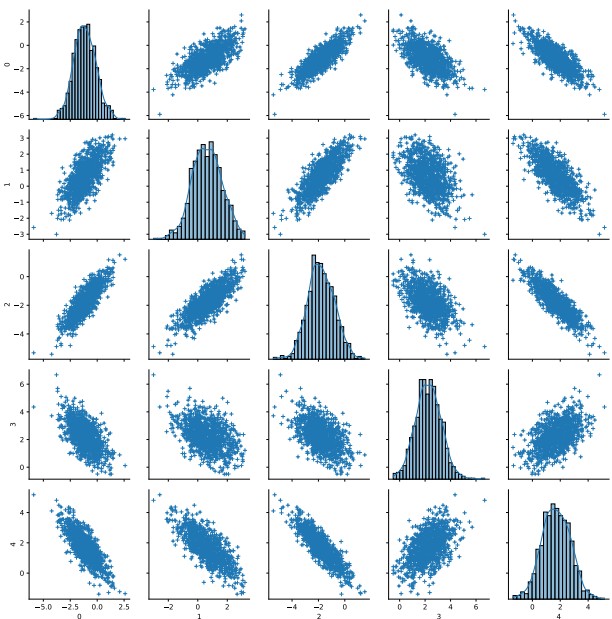

**Figure 4:** Multivariate Gaussian dataset - `mydata1`

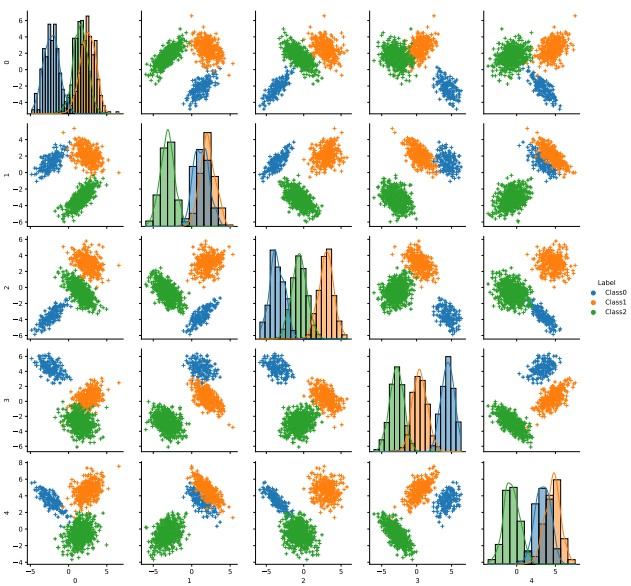

**Figure 5:** Mixture of 3 Gaussians dataset - `mydata2`

## C APPENDIX: HYPERPARAMETERS SELECTION

### C.1 CHOOSING THE NUMBER OF EPOCHS FOR GAIN AND MISGAN

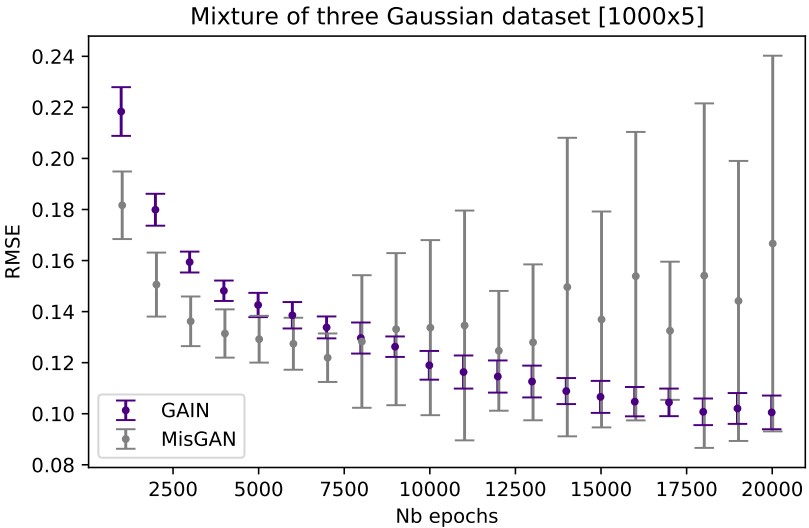

**Figure 6:** Imputation RMSE for both GAN models (MisGAN and GAIN) on `mydata2` dataset in the MCAR scenario with 20% missing rate. We select the optimal number of epochs to be 5,000 for MisGAN and 20,000 for GAIN.

See Fig. 6. While the imputation performances for GAIN are consistently decreasing with a low variance when we increase the number of training epochs, MisGAN shows poor performances with a high variability after 7,000 training epochs (notice the large standard deviations). Consequently, we respectively select 20,000 epochs and 5,000 epochs for GAIN and MisGAN hyperparameters. As the RMSE steadily decreases from 1,000 to 20,000 epochs in the case of GAIN, we decide to investigate more training epochs. See Appendix C.2 below.

### C.2 ADDITIONAL NUMBER OF EPOCHS FOR GAIN

See Fig. 7. If there is a clear improvement from 10,000 to 20,000 epochs, there is no significant change between 20,000 and 100,000 training epochs for GAIN imputation performances. We decide to stick with 20,000 training epochs (per 1,000 observations) for GAIN so that the computational cost does not become prohibitive.

### C.3 CHOOSING THE NUMBER OF NEIGHBOURS FOR KNN

See Fig. 8. Both KNN algorithms have better imputation quality when we increase the number of neighbours from 2 to about 50. KNN-uniform shows better imputation quality than KNN-distance for the generated mixture of Gaussians dataset with MCAR setting at 20% missing rate. KNN-uniform becomes worse than KNN-distance when the number of neighbours becomes to large. Indeed, with the distance weight system, KNN-distance is less sensitive to newly included poor predictors when the number of neighbours becomes unnecessarily large.

Despite the global minimum being around 70 neighbours for KNN-uniform and 120 for KNN-distance, we decide to select $K = 50$ neighbours (per 1,000 observations) as the hyperparameter for both KNN algorithms. The reason behind this choice is that the RMSE plateaus on a large number of neighbours and its standard deviation is too large to justify meticulous fine-tuning on the number of neighbours.

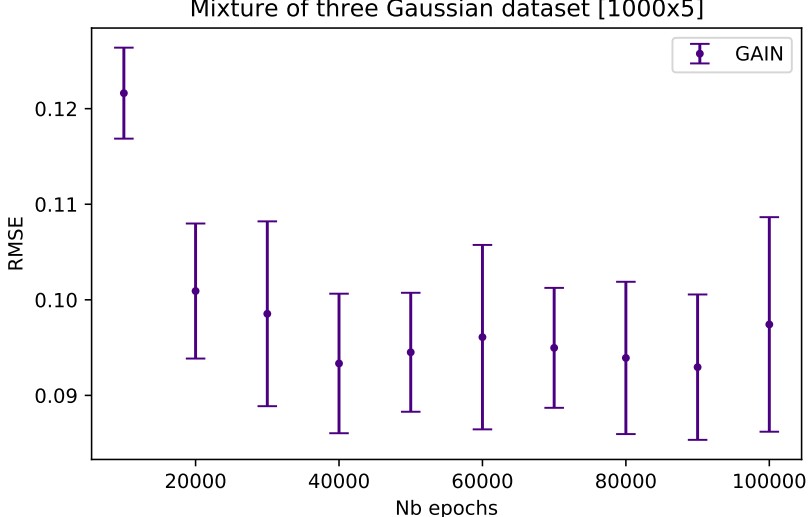

**Figure 7:** Imputation RMSE for additional training epochs with GAIN on `mydata2` dataset in the MCAR scenario with 20% missing rate. There is no significant improvement after 20,000 epochs on the imputation RMSE in spite of a large computation cost increase.

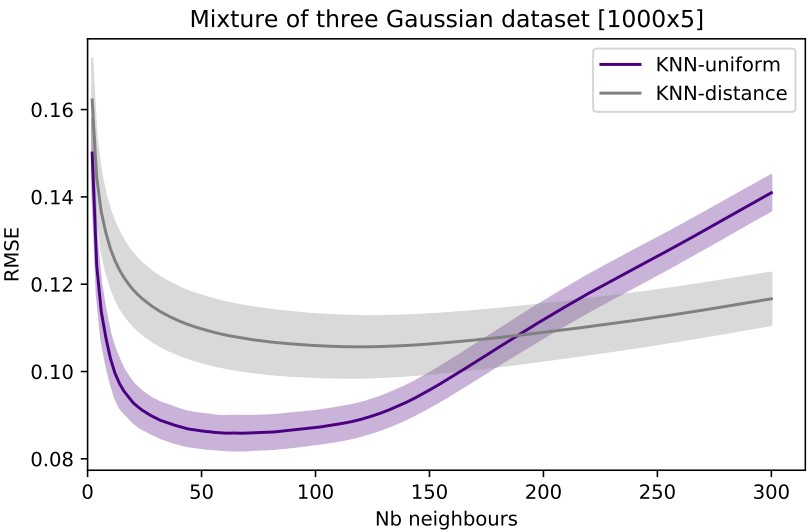

**Figure 8:** Imputation RMSE on `mydata2` dataset in the MCAR scenario with 20% missing rate with the KNN models while varying the number of neighbours. The optimal hyperparameters seems to lie close to N=50 neighbours.

### C.4 VERIFICATION OF THE HYPERPARAMETER SCALING PROCESS

We see in Fig. 9 and 10 the imputation RMSE for GAIN with respect to the training epochs on the left panel, and for both KNN algorithms with respect to the number of neighbours on the right panel. Looking at Fig. 9 corresponding to the `wine_white` dataset first, the scaling process introduced in Section 5.1 recommends to use $\frac{5000}{4898} \times 1000 \approx 1000$ epochs, and $50 \times \frac{4898}{1000} \approx 250$ neighbours. We see that these approximations do not corresponds to the optimal hyperparameters. The optimal hyperparameters correspond to 5000 epochs for GAIN and approximately 25 neighbours for KNN. Now looking at Fig. 10 which uses the `breast` dataset, the recommend values obtained by the scaling process correspond to the optimal values for the algorithms hyperparameters. Indeed, for

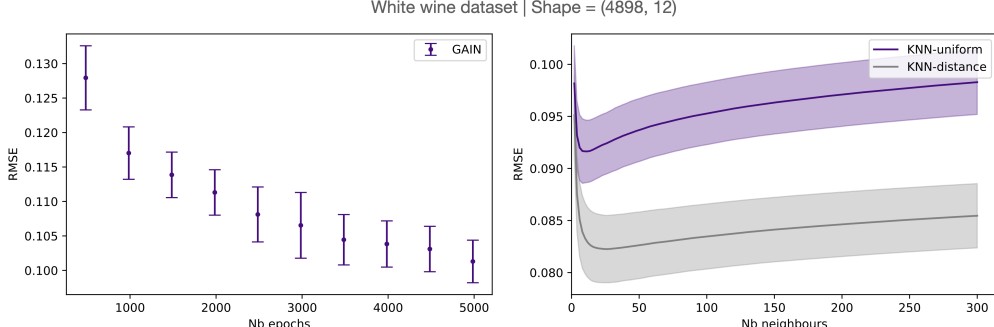

**Figure 9:** Imputation RMSE for the `wine_white` dataset in MCAR scenario with 20% missing rate depending on the hyperparameter selection. The scaling process gives approximately 4000 epochs for GAIN and 250 neighbours for the KNN algorithms.

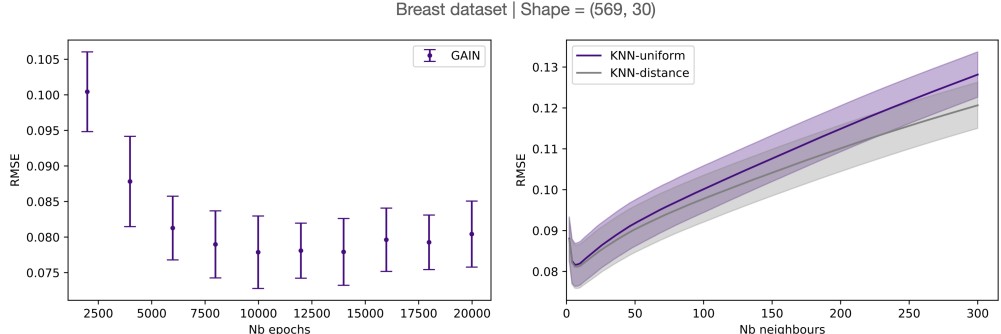

**Figure 10:** Imputation RMSE for the `breast` dataset in MCAR scenario with 20% missing rate depending on the hyperparameter selection. The selected scaling process gives approximately 10,000 epochs for GAIN and 25 neighbours for the KNN algorithms.

GAIN the number of epochs obtained with the scaling process is $frac5000569 \times 1000 \approx 9000$ and for KNN the computed number of epochs is $50 \times \frac{569}{1000} \approx 28$. Looking at the RMSE, the optimal number of epochs for GAIN is 10,000 epochs, and the optimal number of neighbours for the KNN algorithms is 6 (which is lower than our estimate of 28).

Even if we see that the scaling process does not always yield optimal results, we decide to perform the whole analysis of this work following this scaling method. The reason behind this choice is that an extensive hyperparameter search is not only extremely time and energy consuming, but it also impedes unambiguous reproducibility.

## D   APPENDIX: SKIPPED OVER PERFORMANCES

### D.1   POOR IMPUTATION QUALITY WITH MISGAN

Fig. 11 shows the imputation quality of the four algorithms presented in this study (including Mis-GAN), on the mixture of three Gaussians dataset with varying missing rates in MCAR setting. The imputation quality of MisGAN rapidly deteriorates when the missing rate increases. Given the poor performances of MisGAN, we decide not to include any of its results in the study for clarity. This can be explained because MisGAN has been developed as a deep-learning architecture tailored for deteriorated images restoration, while this work focuses on tabular data. We refer the interested reader to Li et al. (2019) for further details on MisGAN. Imputations with MisGAN have been conducted with all experiments of this work (see Fig. 12), but the results are not plotted for clarity.

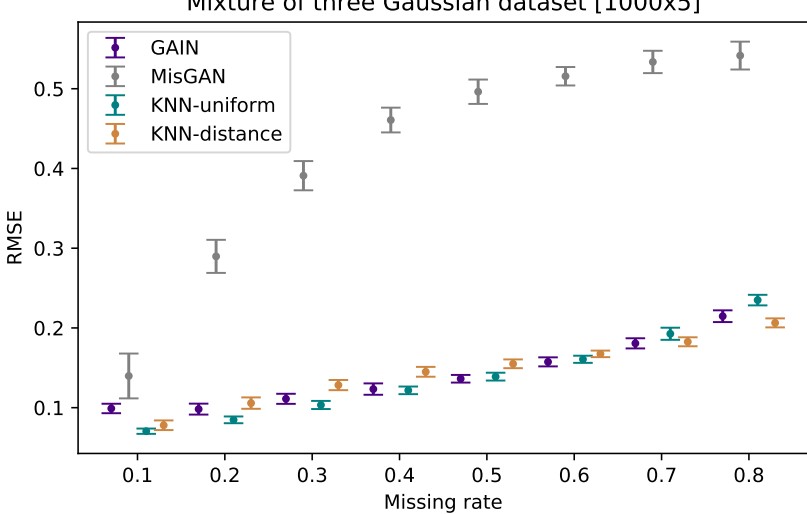

**Figure 11:** Imputation RMSE on `mydata2` dataset in MCAR setting for all data imputation methods with varying missing rates. The performances of MisGAN are particularly poor in comparison to the other three methods.

### D.2 CHAOTIC IMPUTATION PERFORMANCES FOR THE `NEWS` DATASET

Fig. 12 presents the imputation performances of the four algorithms in this study for the seven real-world datasets in MCAR setting with 20% missing rate. We see that MisGAN systematically performs worst, and we therefore decide not to include its results for other experiments.

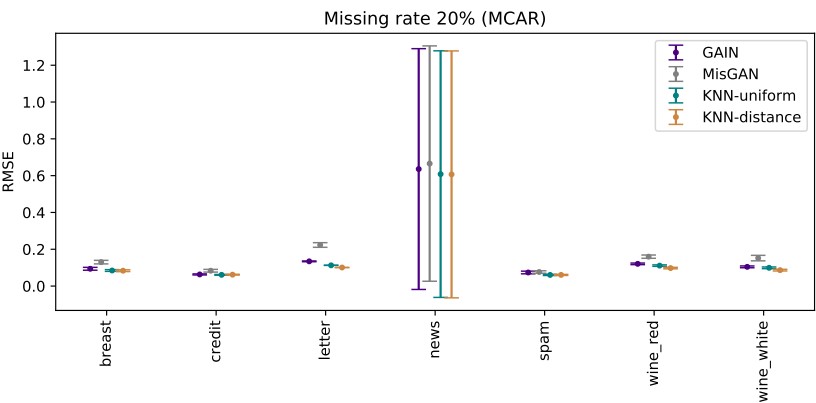

**Figure 12:** Imputation RMSE for all real-world datasets (including the `news` dataset) in the MCAR scenario with 20% missing rate. The performances of the four selected algorithms are extremely variable on the `news` dataset.

We note the huge variability in the imputation performances for the four algorithms over the `news` dataset. The reason is that the `news` dataset has few extreme outliers, whose prediction will be particularly bad when randomly hidden before scaling the remaining data between $[0, 1]$. The same phenomenon happens on the `spam` dataset in the MNAR setting with an average missing rate of 45%: extreme outliers are consequently more likely to become missing values in the hiding process (because of the chosen MNAR setting using quantiles).

We would like to mention that the `news` and `spam` datasets are not really suited for data imputation problems. Using descriptive statistics (not presented in this work), we see that these two datasets are sparse with a lot of zeroes, and have extreme outliers. This is not a problem per se, but we think that

a manual treatment of missing values in such datasets is more appropriate. Unfortunately, they have been extensively used to benchmark Machine Learning algorithms. We decided to include them in this comparative study since they have been used by Yoon et al. (2018).

# E  APPENDIX: QUANTITATIVE ANALYSIS ON THE TRAINING TIME

Recall that the training strategy consists of using 20,000 and 5,000 training epochs per 1,000 observations for GAIN and MisGAN respectively, and using 50 neighbours per 1,000 observations for both KNN algorithms. Table 5 shows the average training time for the algorithms used in this study. KNN-uniform and KNN-distance having really similar training procedures, with grouped them into a common column.

**Table 5:** Average training time per dataset for the four algorithms in this study

| Dataset name | Size | GAIN (in sec.) | MisGAN (in sec.) | KNN (in sec.) |
|:---:|:---:|:---:|:---:|:---:|
| breast | (569, 30) | 221.9 | 88.6 | 0.0 |
| credit | (30000, 14) | 194.3 | 86.9 | 44.8 |
| letter | (20000, 16) | 196.9 | 85.2 | 24.6 |
| news | (39644, 44) | 213.3 | 129.4 | 210.7 |
| spam | (4601, 57) | 220.5 | 89.0 | 3.3 |
| wine_red | (1599, 12) | 200.0 | 82.8 | 0.1 |
| wine_white | (4898, 12) | 197.1 | 82.8 | 1.1 |
| mydata1 | (1000, 5) | 190.5 | 80.4 | 0.2 |
| mydata2 | (1000, 5) | 190.1 | 80.9 | 0.0 |

The training time for GAIN and MisGAN remains nearly constant across datasets. This is because of the multiplicative factor (introduced in Section 5.1) used to keep similar proportion when training the ANN models. To trian GAIN for example, we use 20,000 training epochs for the dataset mydata1 which has 1,000 observations, but we use only 1,000 training for the letter dataset which shows 20,000 observations.

However, the training time for the KNN algorithms is strongly dependent on the dataset size. Indeed, computing the distance matrix between each pair of rows can become computationally expensive for large datasets. For the largest dataset, news, it takes about 3 minutes and 30 seconds to compute the distance matrix. That being said, there is no need to compute the distance matrix again when tuning the hyperparameter $K$ of the number of neighbours used for imputation, while we need to retrain GAIN or MisGAN from scratch if we change the number of training epochs, their hyperparameter in this case.

# F  APPENDIX: RANDOM FACTOR METHOD FOR CORRELATION MATRICES

Generating random correlation matrices can be easily done by transforming a random square matrix $A$ into a symmetric positive definitive matrix. In our case, we draw the entries of $A$ from a standard Gaussian distribution, and we compute $A^T A$ symmetric positive definite matrix. However, this method yields matrices with non-diagonal elements close to zeros.

To obtain a symmetric definite positive matrix of dimension $n$ with higher correlation coefficients outside the diagonal, we start by choosing a number of factors $0 < k < n$. We draw a rectangular matrix $A$ of size $(k, n)$ whose entries are independently sampled from a standard Gaussian distribution. We compute the matrix $A^T A$ which is symmetric of size $(n, n)$, but not full rank. We draw a diagonal matrix $D$ whose entries have random positive noise, such that $W = A^T A + D$ is now invertible.

The code to generate the Gaussian data used is this study is provided as supplementary material.

# G  APPENDIX: NUMERICAL IMPUTATION RMSE RESULTS

**Table 6:** Numerical results corresponding to Fig. 1 (a). The mean and standard deviation of the imputation RMSE are given in percents (%).

| Missing rate | GAIN | KNN-uniform | KNN-distance | MisGAN |
|---|---|---|---|---|
| 10% | 9.9 ($\pm$ 0.59) | 7.05 ($\pm$ 0.33) | 7.8 ($\pm$ 0.61) | 13.98 ($\pm$ 2.81) |
| 20% | 9.82 ($\pm$ 0.69) | 8.47 ($\pm$ 0.42) | 10.57 ($\pm$ 0.71) | 28.98 ($\pm$ 2.07) |
| 30% | 11.11 ($\pm$ 0.63) | 10.34 ($\pm$ 0.51) | 12.83 ($\pm$ 0.64) | 39.09 ($\pm$ 1.83) |
| 40% | 12.33 ($\pm$ 0.71) | 12.17 ($\pm$ 0.48) | 14.5 ($\pm$ 0.62) | 46.07 ($\pm$ 1.55) |
| 50% | 13.63 ($\pm$ 0.48) | 13.89 ($\pm$ 0.49) | 15.5 ($\pm$ 0.55) | 49.62 ($\pm$ 1.53) |
| 60% | 15.74 ($\pm$ 0.57) | 16.07 ($\pm$ 0.45) | 16.75 ($\pm$ 0.41) | 51.57 ($\pm$ 1.16) |
| 70% | 18.07 ($\pm$ 0.64) | 19.27 ($\pm$ 0.76) | 18.26 ($\pm$ 0.56) | 53.36 ($\pm$ 1.4) |
| 80% | 21.47 ($\pm$ 0.74) | 23.49 ($\pm$ 0.66) | 20.63 ($\pm$ 0.56) | 54.16 ($\pm$ 1.75) |

**Table 7:** Numerical results corresponding to Fig. 1 (b). The mean and standard deviation of the imputation RMSE are given in percents (%).

| Dataset name | GAIN | KNN-uniform | KNN-distance | MisGAN |
|---|---|---|---|---|
| breast | 9.39 ($\pm$ 0.77) | 8.46 ($\pm$ 0.47) | 8.35 ($\pm$ 0.47) | 13.01 ($\pm$ 0.96) |
| credit | 6.32 ($\pm$ 0.28) | 6.1 ($\pm$ 0.2) | 6.2 ($\pm$ 0.22) | 8.33 ($\pm$ 0.74) |
| letter | 13.44 ($\pm$ 0.21) | 11.27 ($\pm$ 0.05) | 10.09 ($\pm$ 0.07) | 22.32 ($\pm$ 1.26) |
| news | 63.58 ($\pm$ 65.42) | 60.83 ($\pm$ 67.0) | 60.69 ($\pm$ 67.09) | 66.56 ($\pm$ 63.94) |
| spam | 7.38 ($\pm$ 0.72) | 6.05 ($\pm$ 0.31) | 6.06 ($\pm$ 0.29) | 7.64 ($\pm$ 0.66) |
| wine_red | 12.07 ($\pm$ 0.43) | 11.11 ($\pm$ 0.4) | 9.78 ($\pm$ 0.4) | 15.98 ($\pm$ 0.89) |
| wine_white | 10.44 ($\pm$ 0.49) | 9.92 ($\pm$ 0.46) | 8.65 ($\pm$ 0.45) | 15.2 ($\pm$ 1.49) |

**Table 8:** Numerical results corresponding to Fig. 2 (a). The mean and standard deviation of the imputation RMSE are given in percents (%).

| Dataset name | GAIN | KNN-uniform | KNN-distance | MisGAN |
|---|---|---|---|---|
| breast | 14.35 ($\pm$ 1.14) | 9.63 ($\pm$ 0.88) | 9.49 ($\pm$ 0.87) | 15.66 ($\pm$ 1.11) |
| credit | 5.21 ($\pm$ 0.35) | 5.15 ($\pm$ 0.36) | 5.4 ($\pm$ 0.4) | 7.56 ($\pm$ 0.43) |
| letter | 13.04 ($\pm$ 0.14) | 11.26 ($\pm$ 0.07) | 11.42 ($\pm$ 0.1) | 27.34 ($\pm$ 1.18) |
| news | 77.51 ($\pm$ 68.59) | 75.47 ($\pm$ 70.0) | 75.34 ($\pm$ 70.09) | 82.9 ($\pm$ 65.24) |
| spam | 7.03 ($\pm$ 0.49) | 6.32 ($\pm$ 0.44) | 5.73 ($\pm$ 0.42) | 7.62 ($\pm$ 0.52) |
| wine_red | 12.16 ($\pm$ 0.48) | 11.0 ($\pm$ 0.44) | 9.71 ($\pm$ 0.48) | 18.83 ($\pm$ 1.23) |
| wine_white | 10.43 ($\pm$ 0.41) | 9.86 ($\pm$ 0.43) | 8.89 ($\pm$ 0.43) | 17.22 ($\pm$ 1.05) |
| mydata1 | 9.9 ($\pm$ 0.5) | 9.5 ($\pm$ 0.28) | 10.03 ($\pm$ 0.34) | 21.76 ($\pm$ 1.61) |
| mydata2 | 9.71 ($\pm$ 0.68) | 10.05 ($\pm$ 0.47) | 13.35 ($\pm$ 0.73) | 25.18 ($\pm$ 1.86) |

**Table 9:** Numerical results corresponding to Fig. 2 (b). The mean and standard deviation of the imputation RMSE are given in percents (%).

| Dataset name | GAIN | KNN-uniform | KNN-distance | MisGAN |
|---|---|---|---|---|
| breast | 21.26 ($\pm$ 1.14) | 14.43 ($\pm$ 0.72) | 14.07 ($\pm$ 0.7) | 29.22 ($\pm$ 2.29) |
| credit | 7.02 ($\pm$ 0.44) | 7.15 ($\pm$ 0.36) | 8.06 ($\pm$ 0.44) | 11.48 ($\pm$ 0.91) |
| letter | 14.45 ($\pm$ 0.33) | 12.99 ($\pm$ 0.14) | 13.04 ($\pm$ 0.14) | 31.61 ($\pm$ 1.06) |
| news | 56.19 ($\pm$ 36.05) | 55.07 ($\pm$ 36.81) | 55.55 ($\pm$ 36.48) | 62.73 ($\pm$ 32.42) |
| spam | 9.89 ($\pm$ 0.7) | 7.9 ($\pm$ 0.55) | 9.74 ($\pm$ 0.45) | 8.69 ($\pm$ 0.67) |
| wine_red | 14.83 ($\pm$ 0.94) | 12.97 ($\pm$ 0.49) | 14.6 ($\pm$ 0.58) | 26.91 ($\pm$ 1.12) |
| wine_white | 12.92 ($\pm$ 0.54) | 12.39 ($\pm$ 0.6) | 13.65 ($\pm$ 0.68) | 22.94 ($\pm$ 1.44) |
| mydata1 | 11.6 ($\pm$ 0.47) | 11.25 ($\pm$ 0.38) | 11.78 ($\pm$ 0.34) | 34.02 ($\pm$ 1.82) |
| mydata2 | 14.8 ($\pm$ 0.59) | 17.33 ($\pm$ 0.38) | 19.96 ($\pm$ 0.52) | 37.31 ($\pm$ 1.55) |

**Table 10:** Numerical results corresponding to Fig. 3 (a). The mean and standard deviation of the imputation RMSE are given in percents (%).

| Dataset name | GAIN | KNN-uniform | KNN-distance | MisGAN |
|---|---|---|---|---|
| breast | 14.75 (± 0.98) | 12.14 (± 1.1) | 11.98 (± 1.11) | 20.97 (± 1.69) |
| credit | 7.85 (± 0.44) | 8.1 (± 0.29) | 8.17 (± 0.3) | 11.2 (± 0.73) |
| letter | 13.29 (± 0.18) | 12.83 (± 0.04) | 11.91 (± 0.06) | 29.74 (± 1.08) |
| news | 130.86 (± 60.0) | 130.55 (± 60.37) | 130.5 (± 60.42) | 134.06 (± 57.62) |
| spam | 9.15 (± 0.32) | 8.88 (± 0.37) | 8.01 (± 0.32) | 9.98 (± 0.42) |
| wine_red | 13.18 (± 0.83) | 13.9 (± 0.82) | 12.5 (± 0.82) | 24.33 (± 1.5) |
| wine_white | 11.59 (± 0.37) | 11.86 (± 0.48) | 10.6 (± 0.45) | 22.15 (± 1.67) |
| mydata1 | 9.98 (± 0.54) | 10.5 (± 0.53) | 11.38 (± 0.54) | 27.86 (± 3.4) |
| mydata2 | 9.91 (± 0.56) | 9.99 (± 0.63) | 12.01 (± 0.65) | 27.24 (± 3.06) |

**Table 11:** Numerical results corresponding to Fig. 3 (b). The mean and standard deviation of the imputation RMSE are given in percents (%).

| Dataset name | GAIN | KNN-uniform | KNN-distance | MisGAN |
|---|---|---|---|---|
| breast | 30.3 (± 1.52) | 25.41 (± 1.23) | 25.13 (± 1.24) | 38.35 (± 1.73) |
| credit | 12.14 (± 0.55) | 13.06 (± 0.51) | 13.09 (± 0.51) | 15.99 (± 0.84) |
| letter | 17.36 (± 0.59) | 18.95 (± 0.2) | 19.1 (± 0.21) | 32.23 (± 0.66) |
| news | 158.89 (± 7.9) | 158.97 (± 7.89) | 159.0 (± 7.9) | 161.34 (± 7.86) |
| spam | 23.95 (± 12.63) | 23.2 (± 12.84) | 23.11 (± 12.85) | 23.69 (± 12.68) |
| wine_red | 19.9 (± 1.03) | 23.6 (± 0.97) | 25.02 (± 1.17) | 31.94 (± 1.44) |
| wine_white | 17.16 (± 1.15) | 20.24 (± 1.12) | 21.4 (± 1.18) | 30.16 (± 1.82) |
| mydata1 | 13.69 (± 0.95) | 16.28 (± 0.67) | 17.33 (± 0.77) | 35.03 (± 2.31) |
| mydata2 | 15.04 (± 0.8) | 25.38 (± 1.19) | 24.84 (± 1.03) | 38.8 (± 2.58) |

