# OpenReview forum: "Tabular Data Imputation: Choose KNN over Deep Learning"
_ICLR.cc/2022/Conference — ICLR 2022 Submitted_

### Official Review · Reviewer_VqfN · 2021-10-20

**Correctness:** 3
**Technical Novelty And Significance:** 2
**Empirical Novelty And Significance:** 3
**Recommendation:** 6
**Confidence:** 5

**Main Review:**

Strength:
- The text is clear; the experiments are robust; the messages are important.
- This paper has also an important message for the new method developers within the community suggesting to compare first their new methods with simple standard baselines. The authors might want to emphasize more this point in their summary.
- This study presents a nice guideline for imputing missing values in tabular data.

Weaknesses:
- The experimental design can be improved. Currently, the authors mainly focused on the effect of missing data mechanisms (MCAR, MAR, MNAR) on the performance of data imputation strategies. This is while there are also other important avenues to explore. For example, the authors could also evaluate the effect of different hyperparameter settings on the performance of GAIN and KNN. Currently, we have such analysis in Appendix C but only for simulated data and a fixed number of samples (1000). What about the effect of hyperparameters when imputing real data?  I suggest focussing on one real dataset as a case study and evaluating the effect of hyperparameters (K in KNN and epoch number in GAIN) on the performance of imputation. My hypothesis is that K in KNN is less sensitive to sample size compared to the number of epochs in GAIN. It is also interesting to see that how the optimal value for the hyperparameters changes from one dataset to another. Furthermore, I wonder why the authors decided to include the misGAN in the main text (section 3.2) while its results are below the bar and only presented in the appendix. I suggest saving some space for extra analysis by moving all information about misGAN to appendices.
- The introduction consists few strong (of course valid) arguments and sentences without proper referencing. For example 1) "This solution, although straightforward to implement, has two main disadvantages: it can significantly reduce the size of the dataset and induce a bias [Ref.?]." 2) "It allows to preserve the whole dataset for subsequent analysis but requires careful handling as it can also introduce a bias in the imputed dataset [Ref.?]." 3) "In particular, only ”well-behaved” values can be imputed and missing values of outlier observations will get overlooked [Ref.?]."
- While the argument is to compare the recently introduced complex GAN-based imputation with their simpler classic alternatives, I wonder why the authors did not include the MICE method in their analysis. MICE performs in some ways similar to GAIN and would be interesting to see a comparison. Of course, MICE does not apply to the MNAR scenario (due to its MAR assumption).

Minor comments and questions:
- The last paragraph on page 1 is not informative and can be removed in favor of more space for extra analysis.
- "We argue that all real-world datasets have missing values distributed in MNAR settings; MCAR and MAR are convenient assumptions to facilitate the treatment of incomplete datasets." This is an unsupported argument that should be supported either by some empirical evidence or by including some references.
- Sec 5: the data is scaled to interval [0,1]: is it only for outputs or includes also inputs? Should be clarified in the text because this rescaling might not be the best choice for the input features. I assume this is done only for outputs to make the resulting RMSE comparable across datasets. In that case, again I would suggest skipping rescaling and using instead standardized MSE (SMSE). It is a very minor comment though.
- It is not clear how the other GAIN hyperparameters ($h$ and $\alpha$) are decided. The authors say after several trials they have decided to use the defaults in the original study. Would be nice to include those trials in the appendix.
- Page 3: "parameters are trained trained following..." => "parameters are trained following..."
- Page 3: "After several trails, ..."=> "After several trials,..."
- Some aspects of this study overlap with recent work at https://doi.org/10.3389/fdata.2021.693674 . I wonder why the authors did not mention this study.



**Summary Of The Paper:**

This manuscript presents a nice empirical comparison between the classic KNN imputation with two state-of-the-art GAN-based imputations for tabular data imputation. The experiments are performed on both simulated and publicly available real data. The results, overall, show that the KNN, despite its simplicity, provides very competitive results when compared to its more complex and computationally expensive alternatives. In summary, this paper has a few important messages for both method developers and reviewers in the missing data community: i) first try the simplest; ii) more complex does not mean better; iii) understand your data before data imputation;

**Summary Of The Review:**

This is a nice study with important messages for the community. The experimental section can be enriched with a more in-depth evaluation of the effect of hyperparameters on the performance of data imputation approaches.

---

> ### Author Response · Authors · 2021-11-19
> **Response to Reviewer VqfN**
>
> Thank you very much for your time as well as the valuable feedback.
>
> I have edited the manuscript in light of your feedback. Here I detail the modifications. I start by addressing the three bullet points in the Weaknesses paragraph, and then reply to the seven points in comments and questions.
>
> -- Weaknesses --
> 1) Hyperparameter selection has been done on real-world dataset as suggested. The value of the hyperparameter obtained by exhaustive search has been compared with the one obtained with the scaling process introduced in the manuscript. I have added Appendix C.4. which shows two examples using real-world datasets: one in which the scaling process applies well and another one in which it does not. The reason behind the introduction of the scaling mechanism for hyperparameter setting is that not only an exhaustive hyperparameter search is time and energy consuming, but it also prevents from unambiguous reproducibility.
> 2) A lot of statement were not well supported, indeed. Thank you for pointing this out. I have added relevant references when needed. In particular, several studies already attempted to compare traditional data imputation algorithms with deep-learning architectures.
> 3) The reason why MICE has not been introduced in this study is because it has been introduce in the GAIN study and has been shown to perform relatively poorly. On the contrary, this study aims to focus at a single traditional data imputation algorithm (KNN) against heavy deep-learning algorithms.
>
> -- Minor comments and questions --
> 1) As I have extra space, I prefer to keep the summary paragraph. I personally find it particularly informative when I am reading academic papers.
> 2) The unsupported argument has been rephrased.
> 3) Section 5, both the input and the output of all data imputation algorithms have been scaled in the interval [0, 1]. The reason behind this choice is indeed to have all imputation RMSE comparable across datasets.
> 4) This is an interesting point. Analysis on the hyperparameter hint-rate $h$ for the GAIN algorithm has not shown any impact. But little has been thoroughly investigated for $h$.
> For $\alpha$ however, I have found that it has a binary behaviour: either $\alpha > 0$ in which case the loss of the generator will first try to minimize the RMSE (used for the fit to the non-hidden data) before entering the adversarial process with the discriminator ; or $\alpha = 0$ in which case the loss of the generator does not take the RMSE into account (only the adversarial process takes place in GAIN). The magnitude of $\alpha$ did not appear to play a role at all. Therefore, I decided to fix the value of this hyperparameter to $\alpha = 100.0$ as the author of GAIN recommended to.
> 5) Fixed
> 6) Fixed
> 7) Thank you for the references, I have added it to the manuscript and edited the abstract, introduction and conclusion in this regard.
>
> Again thank you for your valuable input. I hope my edits and this response correctly address your questions and comments.
> I remain at your disposal for further discussion.
> Best,

---

### Official Review · Reviewer_zJAq · 2021-10-28

**Correctness:** 2
**Technical Novelty And Significance:** 2
**Empirical Novelty And Significance:** 2
**Recommendation:** 3
**Confidence:** 5

**Main Review:**

**List strong and weak points of the paper. Be as comprehensive as possible.**

**Pros:**

* All missingness scenarios (MCAR, MAR, MNAR) are taken into account.
* The (real-world) datasets' shapes range from 12 to 57 attributes and 569 to 39644 examples and, therefore, cover a wide range of scenarios.
* All datasets are fully-observed and the missing values are artificially induced. This helps to eliminate interfering effects.
* Structure of the paper is clear

**Cons:**

* The paper claims to be "the first to compare state-of-the-art deep-learning models with the well-established KNN algorithm", which seems like a bit of a bold statement given the number of publications on empirical work in that space in the past years:
  * papers that present imputation methods sometimes use KNN as baseline (see: <https://arxiv.org/pdf/1808.01684.pdf>)
  * there is a number of papers at a recent ICML workshop (see <https://artemiss-workshop.github.io/>), where several papers presented empirical work on KNN imputations as well as more recent generative deep learning approaches
  * there there is recent benchmark that compares both: <https://www.frontiersin.org/articles/10.3389/fdata.2021.693674/full>
  * there is a recent paper comparing deep learning approaches with AutoML imputations, and KNN is one of the models used in that AutoML solution <https://arxiv.org/abs/2106.11189> - interestingly that publication finds that if regularized properly, deep nets can actually outperform more classical discriminative methods (such as KNN)
* The algorithms hyperparameters are optimized on a synthetic dataset and then “scaled” to the dataset’s size. Hyperparameters should always be tuned on the same that that the models are applied to. In the very setting investigated by the authors, it is fair to assume that KNN, with fewer hyperparameters, has an advantage due to that procedure
* an important experimental setting is whether the imputation model was trained on fully observed data (a relevant but rather academic/lab-conditions setting) or on data with missing values (which is more relevant in real-world applications).
* Minor:
  * Section 7.2 abbreviation "CNN" should be introduced


**Clearly state your recommendation (accept or reject) with one or two key reasons for this choice.**

I vote for reject since there are already comparisons between KNN and state-of-the-art models. Further, the hyperparameter optimization can explain the better performance of KNN.

**Ask questions you would like answered by the authors to help you clarify your understanding of the paper and provide the additional evidence you need to be confident in your assessment.**

* Although, the method used to induce MAR missingness is clear to me, I do not understand the column “MAR column” in Table 1.
* Is there a good reason why experiment 1+2 differ from the third and fourth?

**Provide additional feedback with the aim to improve the paper. Make it clear that these points are here to help, and not necessarily part of your decision assessment.**

* Given an overview of the datasets' attributes data types would help to get more insights. Perhaps you should consider to take them into account for the discussion.
* Add information about the data split or cross-validation used to find hyperparameters would increase the credibility.
* Figures 3+4 and 5+6 could be side-by-side with the same y-axis; in my opinion, figures 1+2 too to support readability.
* I personally prefere extensive figure captions.
* Lastly, adding more "simple" imputation methods, e.g. random forests, neural networks, would be interesting, since there are some empirical evaluations that come to different results
* the authors are right in that there are few peer reviewed papers at the big ML conferences that directly compare latest generative deep learninig methods with classical methods, like kNN. This is however in part true because in my experience these papers do not get accepted. I have been reviewing imputation papers for the past couple years and I've seen several cases where an empirical evaluation of this kind, even though the experimental setting was absolutely flawless, did not make it.

**Summary Of The Paper:**

The paper compares two generative deep learning based data imputation methods (GAIN and MisGAIN) with KNN based imputation. The authors conducted experiments on seven real-world and two synthetic datasets in  all missingness scenarios (MCAR, MAR, MNAR) and multiple missingness rates.

**Summary Of The Review:**

The experimental setting has a significant limitation, the hyper parameter optimization is not conducted on the right data set, which can explain the superior performance of KNN. The authors claim that they are the first to report these results, which is not exactly true.

---

> ### Author Response · Authors · 2021-11-19
> **Response to Reviewer zJAq**
>
> Thank you for your time and detailed feedback.
>
> I have added corrections to my manuscript based on your recommendations, and would like to share the details here. I start by answering to the 4 points in "Cons", then the 2 questions, and finally the 6 additional feedback bullet points.
>
> -- Cons --
> 1) Indeed, a lot of interesting literature has been overlook when writing the manuscript. Thank you very much for your references. I have added them into to manuscript and rewritten the abstract, introduction and conclusion parts.
> 2) Regarding the scaling of the hyperparameters, I have added Appendix C.4, in which I show in which situations the scaling process applies correctly to the hyperparameter settings. The reason behind this choice is that an exhaustive optimization for the hyperparameter on every dataset in time and energy consuming, as well as complex to reproduce.
> 3) The data imputation models have always been trained on dataset with injected missing data. Never did I trained a model on a complete dataset before imputation. The reason behind this choice is that it better reflects real life scenarios.
> 4) The acronym CNN has been removed, thank you for pointing this out.
>
> -- Questions --
> 1) To introduce missing data in a MAR scenario, the "MAR column" presented in Table 1 has been kept untouched. I have used the quantiles of the selected column to define missing probabilities for all other columns. Since the "MAR column" has been kept untouched (no missing data), the resulting dataset is an example of MAR dataset.
> 2) Experiment 1 consists in varying missing rates in the MCAR scenario. Experiment 2 uses only a missing rate of 20% for all real-world datasets. Experiments 3 and 4 both use two missing rates (20% and 45%) in their respective missing data scenarios (MAR and MNAR). Experiment 1 focuses on a single dataset and investigates more missing rates than I do in Experiments 3 or 4.
>
> -- Additional feedback --
> 1) I have added an exhaustive overview of the datasets in Appendix B. I have also modified the discussion to take into account the fact that data imputation performances depend a lot on the nature of the datasets at play.
> 2) I have modified Section 5.1 on hyperparameter selection, and added sentences on the process for imputation RMSE calculation during the hyperparameters search.
> 3) Thank you for this advice! I have merged figures 1 and 2, 3 and 4, 5 and 6.
> 4) Extensive captions have been added to all figures.
> 5) The reason why this study only focuses on the KNN is that we have evidence that the KNN performs best for data imputations amongst traditional data imputation algorithms. I wanted this study to be straightforward and only compare the best "traditional" data imputation scheme versus the best "deep-learning generative models" architecture.
> 6) Thank you for letting me know.
>
> Again thank you for the valuable input. I hope the revisions added to the manuscript answers your questions and comments.
> I remain at your disposal for further discussion.
> Best,

---

### Official Review · Reviewer_m2kd · 2021-10-30

**Correctness:** 2
**Technical Novelty And Significance:** 2
**Empirical Novelty And Significance:** 3
**Recommendation:** 3
**Confidence:** 4

**Main Review:**

This paper considers the problem of data imputation, which is important in handling tabular data, and shows interesting results regarding imputation methods, with some potential practical implications regarding the efficient methods to use. However, there are some concerns regarding the presented results:
- The authors state that they did not manage to reproduce the results of GAIN from the original paper, although they used the same datasets (in one case it was much better in their measurement, in the others worse). They also do not provide reference results for any of the other classical algorithms that appeared in the original paper (such as MICE and MissForest). It is therefore unclear whether their setting was correct and whether this comparison was valid.
- Since datasets behave differently, it is unclear whether setting the hyperparameters based on one small dataset is the right approach. Note that GAIN is better than KNN on this dataset (my_data2), possibly due to some over-fitting, but not as good on the other datasets. It may be better to set the hyperparameters based on several datasets with different mechanisms of missing data. Maybe there could also be some fine-tuning on a subset of the relevant dataset (although it has missing data), rather than assuming a linear scaling based on the number of samples. Also, it should be noted that more hyperparameter options were checked for KNN than for GAIN.
- The difference between the algorithms is shown in figures, but it is not quantified. The paper should state what is the average difference between GAIN and each of the KNN methods, per dataset, per scenario, and overall.
- The paper uses a small number of datasets, and all the datasets used have a small number of features. KNN is known to suffer from the "curse of dimensionality". It would therefore be interesting to see if this result holds for a couple of datasets with hundreds of features or more. It would also be interesting to see the results for a dataset with a larger number of samples.



**Summary Of The Paper:**

This paper considers the problem of tabular data imputation to handle missing numeric data. It compares the RMSE of recently proposed GAN-based methods (GAIN and mis-GAN) with the classical KNN method (either through uniform or distance-weighted imputation). It considers 9 datasets, two are simulated by a mixture of Gaussians and 7 are real datasets, 5 of them used in the GAIN paper. The datasets have 500-30000 samples and 5-57 features (most are below 20). It also considers three different mechanisms of missing data (MCAR - completely random, MAR - random based on the observed data, MNAR - based on unobserved data), each with one or two rates of missing data (more rates are tested for MCAR with the simulated data). The hyperparameters of all the models are tuned once on the simulated data (1000 samples), and scaled based on the number of samples for each dataset (scaling the number of neighbors for KNN, or epochs for the GANs). The models are applied to each imputation case 20 times, presenting the mean and standard deviation for each dataset in each scenario. The paper shows that mis-GAN, originally developed for filling missing image pixels, usually performs much worse than KNN. Furthermore, KNN generally performs roughly on-par compared to GAIN, depending on the dataset and exact scenario (in many cases KNN is better). Based on the presented figures, the RMSE differences are usually within 0.02, but may reach 0.05. The compute required by KNN is much lower, and the paper calls for using the cheaper option, avoiding waste of resources.

**Summary Of The Review:**

The paper considers an interesting problem and presents interesting results that encourage trying KNN. However, due to the concerns mentioned above regarding the correctness and representativeness of the results, and since it does not have a significant technical contribution other than comparing existing methods, it is below the bar for ICLR.

---

> ### Author Response · Authors · 2021-11-19
> **Response to Reviewer m2kd**
>
> Thank you very much for your time and valuable feedback.
>
> I have added corrections to the manuscript based on your comments:
>
> 1) Regarding GAIN results that I could not reproduce, I have provided my code as supplementary materials. I have implemented a personal version of GAIN that I use for this study. The comparison with other algorithms (such as MICE or MissForest) has not been done since there is evidence to support that KNN performs best amongst traditional algorithms for tabular data imputation.
>
> 2) I agree that the hyperparameter setting is the weak point of this study. I have conducted extra analysis and performed the hyperparameter selection on other datasets: results are presented in Appendix C.4.
> This is true that more hyperparameters are tested for KNN than for GAIN. But I believe that the larger number of trials does not mean that KNN has an advantage over GAIN. The imputation RMSE provided in Appendix C is a rather smooth function of the hyperparameter, and a very precise tuning of the hyperparameter is not necessary because of the high variability of the imputation RMSE regardless of the method.
>
> 3) Thank you for pointing this out.
> I have added Appendix G with all numerical values (mean and standard deviation for all experiments).
>
> 4) Indeed, KNN suffers from the curse of dimensionality. Its complexity in O(n^2 d^2) makes it computationally expensive for large datasets.
> That said, it remains computationally faster than deep learning methods even with the larger dataset used in this study (size 39644x44).
>
> I hope my corrections could answer your points. I remain at your disposal for further discussion.
> Best wishes,

---

### Decision · Program_Chairs · 2022-01-20

**Decision:**

Reject

**Comment:**

This paper presents a study of methods for tabular data imputation. In particular, the authors compare deep learning methods with k-NN based approaches. Experiment results demonstrate k-NN to be competitive with deep learning methods.

Reviewers have concerns about the characteristics of datasets used in the experiments and hyperparameters used for evaluation, which I agree with. They also think that the main contribution of comparing k-NN and deep learning methods is not strong enough for acceptance to ICLR. I recommend rejecting this paper.